# Oral Health Behaviour, Care Utilisation, and Barriers among Students with Disabilities: A Parental Perspective

**DOI:** 10.3390/healthcare12191955

**Published:** 2024-09-30

**Authors:** Faris Yahya I. Asiri, Marc Tennant, Estie Kruger

**Affiliations:** 1Department of Preventive Dental Sciences, College of Dentistry, King Faisal University, Al-Ahsa 31982, Saudi Arabia; 2International Research Collaboration—Oral Health and Equity, School of Allied Health, The University of Western Australia, Perth, WA 6009, Australia; marc.tennant@uwa.edu.au (M.T.); estie.kruger@uwa.edu.au (E.K.)

**Keywords:** oral health, disabilities, dental care utilisation, barriers to dental care, parental perspective, Saudi Arabia

## Abstract

Background: Oral health is a critical component of overall well-being. However, students with disabilities (SWDs) face unique challenges in maintaining oral hygiene and accessing dental care. This study aims to explore oral health behaviours, dental care utilisation, and barriers to accessing dental care among SWDs in Al-Ahsa, Saudi Arabia, from the perspectives of their parents. Methods: A descriptive cross-sectional study was conducted among parents of SWDs in Al-Ahsa, Saudi Arabia. Data were collected using a validated questionnaire covering oral health behaviours, dental care utilisation, and barriers to care. The sample size was determined based on the SWDs population in the region. Data were analysed using descriptive and analytical statistics, with significant associations identified at *p* < 0.05. Results: Findings revealed that 45.7% of SWDs brushed irregularly, with only 16.8% brushing twice a day or more. Dental flossing was reported by just 4.3% of SWDs. Emergency-based dental visits were common (51.9%), and 29.5% of SWDs had never visited a dentist. Significant associations were found between maternal education and tooth brushing frequency (*p* = 0.004) and between parental education and the frequency of dental visits (*p* = 0.035). The most reported barriers included fear of treatment (47.1%), difficulty finding willing dentists (45.5%), and long wait times for appointments (44.7%). Conclusions: The study emphasises the pressing need for targeted interventions to improve oral hygiene practices and enhance accessibility to dental services for SWDs.

## 1. Introduction

The United Nations defines disabilities as long-term impairments—whether physical, mental, intellectual or sensory—that, when combined with social and environmental barriers, can restrict an individual’s full and equal participation in society [1]. The World Health Organization (WHO) has further explained that disabilities arise from the interplay between health conditions and various personal and environmental factors [2]. Children and young people with special educational needs and disabilities often face significant challenges in maintaining good oral hygiene. These challenges include reliance on inadequately trained parents or caregivers, limited mobility, oral aversion, clenching or biting on the toothbrush and hypersensitivity to oral stimulation [3]. Oral health is an integral part of overall health, with significant impacts on individuals’ well-being and daily functioning. The burden of oral diseases is immense, affecting almost 3.5 billion people worldwide, with untreated dental caries being the most prevalent condition, impacting around 2 billion individuals [4]. Untreated dental caries can severely affect their ability to eat, speak, learn, and sleep, leading to pain, infection, and reduced school attendance. These negative impacts not only hinder a child’s educational performance but also contribute to broader social inequalities, disproportionately affecting those from disadvantaged backgrounds [4]. Persons with disabilities (PWDs) are at greater risk for poor oral health, and they face greater challenges in accessing health care [2]. Previous studies have shown that PWDs often face barriers to accessing and utilising oral health care, leading to poorer oral health outcomes and higher rates of unmet dental needs. These studies identify challenges across a wide range of disabilities, including physical, sensory, intellectual, autism spectrum disorder, and multiple disabilities. Commonly identified barriers—including availability of appropriate services, accessibility, affordability, and awareness—indicate that disabilities alone are not the sole cause of oral health disparities among PWDs [5,6,7,8,9]. Previous studies have highlighted the impact of disability on oral health and access to care, often focusing on disability-related factors [5,6]. However, there is a need to adopt a broader perspective beyond the biomedical model, considering multi-level barriers—including individual, professional, societal, and systemic barriers—that influence oral health care for PWDs [5,6]. In Saudi Arabia, the prevalence of persons with disabilities is a notable public health issue, with roughly one out of every thirty citizens experiencing some form of disability [10]. Previous research from Saudi Arabia has shown that PWDs, including students, have poorer oral health outcomes than their peers without disabilities [11,12]. This disparity in oral health status underscores the importance of understanding the specific challenges faced by this population. Caregivers play a critical role in this context by making decisions on behalf of the care recipient and managing their interactions with health services, and as supporters by assisting with oral care routines and facilitating dental visits [13,14].

Therefore, this study aims to explore oral health behaviours, dental care utilisation, and barriers to accessing dental care among students with disabilities (SWDs) in Al-Ahsa, Saudi Arabia, as reported by their parents. The study provides a descriptive cross-sectional analysis, examining how sociodemographic factors influence oral health practices and access to care among these students.

## 2. Methodology

### 2.1. Study Design and Setting

This descriptive cross-sectional study was conducted across 21 schools in Al-Ahsa, Saudi Arabia, providing education for SWDs. The study received approval from the Human Research Ethics Committee of the University of Western Australia (file reference—2022/ET000328 on 7 September 2022) and the Research Ethics Committee at King Faisal University (approval number KFU-REC-2022-APR-EA000553 on 5 April 2022), ensuring adherence to ethical guidelines. We obtained informed consent from all participants before they completed the questionnaire. We maintained participant confidentiality and data anonymisation throughout the study, with documents deidentified using codes and removing all identifying information, such as school names. The Planning and Development Department at the General Directorate of Education in Al-Ahsa, Saudi Arabia, granted permission to conduct the study at designated educational institutions.

#### 2.1.1. Sampling

We employed a decentralised, convenience sampling technique to select participants, focusing on their accessibility and willingness to participate. Eligible participants for this study were students with disabilities officially registered as having disabilities within the education system in Al-Ahsa. The students ranged in age from 6 to 22 years, including children and young adults, with a variety of disabilities, including sensory disabilities (such as hearing or vision impairment), cognitive disabilities, autism spectrum disorder (ASD), and others with multiple impairments. All participants were enrolled in special education programs. Only students whose parents provided consent to participate were included in the study. Students without parental consent or who were not officially registered as having disabilities were excluded.

#### 2.1.2. Sample Size

We used an online calculator (Raosoft, http://www.raosoft.com/samplesize.html, accessed on 8 January 2023) to determine an appropriate sample size, aiming for an absolute precision of 5% and a 95% confidence level. The calculation was based on the expected population of SWDs in Al-Ahsa, Saudi Arabia, estimated at 4344, according to the most recent data from Saudi Arabia’s Ministry of Education [15]. This resulted in a minimum required sample size of 354.

### 2.2. Instrumentation

We obtained informed consent from all participants. We aligned with the general objectives of our research [16]. While the original survey focused primarily on barriers to accessing dental care, we expanded it to cover a broader range of variables critical to our study. Specifically, we rearranged the original questions and added new items to capture essential oral health behaviours, such as frequency of tooth brushing and the use of dental floss, as well as detailed patterns of dental service utilisation, including the frequency of dental visits and types of dental procedures received. This process involved seeking expert input to ensure the relevance and comprehensiveness of the questionnaire.

To validate the modified questionnaire, we conducted pilot testing. Initially, the survey was piloted with five male and five female parents who provided feedback on item inclusion, wording clarity, and overall structure. Based on their feedback, we made adjustments to enhance the questionnaire’s effectiveness. We conducted a further pilot test with 10 parents to ensure the final version’s validity and clarity. The barrier scale, which included eight items, demonstrated a reliability coefficient of 0.79, as assessed using Cronbach’s alpha. Respondents ranked several barriers on a scale from 1 to 10, with 10 indicating the highest level of difficulty, following a procedure similar to the original survey [16]. We also used this scale to assess parental attitudes toward routine dental visits and satisfaction with dental care. The full version of the questionnaire used in this study is available in Appendix A.

### 2.3. Data Collection

Data collection took place from February to June 2023. Each school independently identified eligible students with the assistance of teachers and staff from the Special Education Department and the School Health Affairs Department. We distributed 754 paper invitations to participate, including a consent form, study information, and questionnaire to eligible parents of SWDs attending special schools. Parents received reminders to encourage survey completion. Parents who agreed to participate signed the consent forms, completed the questionnaires, and gave them through SWDs, which were later collected by their schools. Parents were also informed that they could contact the lead researcher by email or arrange a meeting to discuss the study and address any questions. All eligible participants received a toothbrush and toothpaste to promote consistent brushing habits.

### 2.4. Statistical Analysis

We conducted data analysis using SPSS version 25.0 (IBM Corp, Armonk, NY, USA). We used descriptive statistics, including frequencies and percentages, to summarise sociodemographic variables, oral health behaviours, oral health utilisation, barriers to dental care, and parental attitudes toward and satisfaction with dental care. Chi-square tests determined associations among these factors, with statistical significance set at *p* < 0.05.

## 3. Results

The study achieved a response rate of 49.9%, with 376 parents successfully completing the survey. The study included mothers (52.1%), fathers (40.2%), and other caregivers such as sisters or grandmothers (7.7%). The distribution of student ages showed that 55.9% were ages 6–12, 33.8% were ages 13–17, and 10.4% were ages 18–22. Most mothers had education levels above high school (35.9%), followed by high school education levels (29.3%). Fathers also predominantly had education levels above high school (27.7%) or high school education levels (28.7%). Most mothers were unemployed (83.5%), whereas most fathers were employed (75.8%; Table 1).

Regarding hand coordination and the ability to brush teeth, 71.0% of students could brush without limitations, whereas 18.6% could do so with some limitations. Regular tooth brushing (twice a day or more) was reported by 16.8% of SWDs, whereas 45.7% brushed irregularly. Dental flossing was uncommon, with only 4.3% reporting regular use. Sugar/sweets intake varied, with 48.1% of students in the study reporting consuming sweets at least once or twice a day in small quantities. A notable portion (28.5%) reported rarely consuming sweets daily, and when they did, it was in small quantities, whereas only 3.5% of the students never consumed sweets. However, 19.9% of the students consumed sweets frequently in large quantities, either once or twice a day or more than twice a day. Perceived oral health status was reported as excellent by only 14.4%, good by 40.1% and moderate to poor by 45.5% of participants (Table 2).

Dental visits were generally infrequent, with 51.9% visiting the dentist only on an emergency basis and 29.5% never visiting at all. Tooth extractions (29.3%) and fillings (25.8%) were the most common dental procedures performed. Notably, only 27.4% of students received sedation for dental treatment. Among those who received sedation, the majority (41.0%) were treated under general anaesthesia in a hospital setting, followed by 20.3% who received oral medication for sedation, 13.5% who received minimal inhaled sedation (‘laughing gas’) and 2.9% who underwent IV conscious sedation. Additionally, 22.3% of the parents were unsure or could not remember the type of sedation used. The type of facility used for dental care varied, with hospitals being the most frequently visited (23.1%), followed closely by paediatric dental practices (22.3%; Table 3).

Parents identified several barriers to dental care, with the most significant being fear of dental treatment (47.1%) and difficulty finding a dentist willing to treat their child (45.5%). This was closely followed by appointment wait times, which 44.7% of parents also rated as difficult. Financial barriers were another notable concern for 37.0% of parents. In contrast, transportation and distance were less problematic, with 42.0% of parents rating it as easy, while distance was somewhat difficult for 44.7% of parents (Figure 1).

Caregivers’ satisfaction with current dental care indicated that only 37.0% were satisfied, whereas 34.8% were somewhat satisfied, and 28.2% were not satisfied (Figure 2). The importance of access to routine dental care was highly valued, with 72.9% considering it important, whereas only 10.1% deemed it unimportant (Figure 3).

We further analysed parents’ demographics to examine associations with the other reported variables. Statistically significant associations were found between maternal education and the frequency of toothbrushing (*p* = 0.004), with higher educational levels correlating with better oral hygiene practices. The relationship to the child also impacted the frequency of dental flossing (*p* = 0.012), with mothers more likely to report SWDs flossing. Sweets intake frequency significantly correlated with both maternal (*p* = 0.036) and paternal education levels (*p* = 0.04; Table 4). Furthermore, we found no statistically significant associations between parents’ perceived oral health status of SWDs and their demographics.

Table 5 shows the associations among parents’ demographics and oral health care utilisation variables. Fathers’ education level statistically and significantly correlated with the frequency of dental visits (*p* = 0.035), as did paternal occupation (*p* = 0.038). Higher educational levels among fathers and paternal employment status both correlated with more frequent dental visits.

We found no statistically significant associations among sociodemographic factors and barriers to dental care, except for financial barriers and interior design barriers. Families with employed fathers were more likely to rate financial barriers as less difficult (*p* = 0.008). However, 66.91% of these families still considered financial barriers difficult, indicating that stable employment does not fully mitigate financial challenges to accessing dental care. Additionally, a higher percentage of mothers with higher education levels tended to report financial barriers as easier to manage compared to those with less education (*p* = 0.037), although a notable portion of these mothers still found financial barriers somewhat difficult (Table 6).

Mothers as primary caregivers reported the highest difficulty with interior design barriers in dental facilities, with a statistically significant association (*p* = 0.048). Fathers with higher education levels were somewhat less likely to rate interior design barriers as difficult, although challenges persisted even among this group (Table 6).

Paternal education was statistically and significantly associated with perceived importance of routine dental care (*p* = 0.041). Other sociodemographic factors, such as the relationship to the child, maternal education, and parental occupation, did not show significant impacts on the perceived importance of routine dental care (Table 7). Additionally, regarding parents’ satisfaction with current dental services for SWDs, there was no association between satisfaction with current dental care and sociodemographic factors.

## 4. Discussion

This study is among the first to comprehensively examine oral health behaviours, dental care utilisation, and barriers to accessing dental services among SWDs from the perspectives of their families. Despite a majority of parents reporting that SWDs were capable of brushing without limitations, irregular tooth brushing and low use of dental floss were common. These findings suggest that while physical capability exists, there may be gaps in oral health education or support, particularly in home environments. These results are consistent with other studies conducted in various regions of Saudi Arabia that have reported similar poor oral health practices among SWDs [12]. The statistically significant association between higher maternal education and more frequent tooth brushing suggests that a mother’s education level may play a crucial role in shaping oral hygiene practices. This aligns with a cross-sectional study from China involving 8446 families, which found that higher maternal education was significantly linked to more frequent tooth brushing among SWDs [17]. Similarly, a Romanian study of 814 schoolchildren showed a positive correlation between mothers’ education levels and increased tooth brushing frequency [18]. These findings consistently highlight the crucial role of maternal education in shaping children’s oral hygiene practices. However, the persistence of irregular brushing even among SWDs of more educated parents suggests that other factors, such as time constraints or lack of parental supervision, may also contribute. Our study also found a statistically significant association between the relationship to the SWDs and flossing, with mothers being more likely to report that SWDs floss. Despite this, the overall rate of flossing among SWDs remains very low (4.3%). This finding highlights that while mothers might be more involved in promoting flossing, the practice itself is rarely adopted.

Furthermore, this study found a notable association between daily sugar/sweets intake and parents’ education levels, particularly among those with education above high school. students of more educated parents were more likely to either never consume sweets or consume them rarely and in small quantities. However, it is also notable that some SWDs of more educated parents still engage in daily sweets consumption, whether in small or large quantities. This underscores the need for tailored and evidence-based approaches to improve oral health behaviours, particularly in the context of parental education and its influence on students’ oral health. Cochrane reviews have consistently highlighted the importance of such interventions. For instance, Riggs et al. (2019) emphasised the role of educational interventions that promote evidence-based nutritional guidelines, starting as early as pregnancy. These interventions are crucial for fostering healthier dietary habits and reducing the risk of early childhood caries by minimising the intake of sugary foods that contribute to dental problems [19]. Zhou et al. (2020) further supported the necessity of specific, evidence-based strategies to address unique challenges in maintaining oral hygiene, suggesting that comprehensive training for both caregivers and people with disabilities is crucial for improving oral health behaviours across diverse populations [20]. This could be facilitated within the school environment, where teachers play a pivotal role in reinforcing comprehensive oral health promotion, including structured oral health education, supervised toothbrushing programmes, and restricting the use of sugary foods. This approach is more effective when teachers have the resources and collaborate with parents and oral healthcare providers, as demonstrated in a recent study in the area [21].

Although a majority of parents acknowledged the importance of routine dental care, many SWDs still relied mostly on emergency visits, reflecting a reactive rather than preventive approach to oral health care. This is consistent with findings from a systematic review by Bhadauria et al. (2024) which identified similar patterns of care among individuals with disabilities, whereby dental services were primarily accessed during emergencies because of barriers to accessing regular care [22]. This disconnect between awareness and practice, along with lower satisfaction levels, raises concerns about accessibility. Furthermore, while oral health providers in the study area are predominantly general dental practitioners [23], the finding that primary health services were the least utilised in this study (with only 16.5% of participants reporting their use) further emphasises these concerns. Most dental care for people with disabilities is not inherently complex and can be effectively delivered in primary care and community settings, provided that dental healthcare professionals are properly prepared and have the right attitudes [24,25]

The statistically significant associations between paternal education (*p* = 0.035) and paternal employment (*p* = 0.038) with more frequent dental visits highlight the role of socioeconomic factors in access to care. Similarly, a recent scoping review by Fadila et al. (2024) emphasised the influence of these sociodemographic factors on dental visit patterns. However, the continued reliance on emergency visits, even among those with higher socioeconomic status, points to broader issues to be addressed [26].

Barriers to dental care, particularly fear of treatment, wait time for dental appointments, and difficulties in finding willing dentists, were the most frequently reported. These challenges highlight the need for more dental professionals trained in managing SWDs and the development of more welcoming dental environments. Enhancing dental environments with sensory adaptations, such as dimmed lighting, calming music, weighted blankets, and tactile familiarisation, as well as assistive technologies like hearing aids, sign language interpreters, and braille materials, can significantly reduce anxiety and improve cooperation during dental visits in persons with disabilities, including SWDs [27,28]. Despite the benefits of these adaptations, there remains a critical need for education and training for both current and future dentists to enhance their preparedness and attitudes in managing SWDs, which could ultimately lead to improvements in dental care access for SWDs [29,30].

Although financial barriers were less frequently cited compared to these challenges, they remain a significant concern, particularly for parents without stable employment. The statistically significant association between financial barriers and paternal employment (*p* = 0.008) indicates that while employment may reduce financial challenges, it does not eliminate them, as a notable percentage of employed fathers still reported financial difficulties. Similarly, the statistically significant association between mothers with higher education levels and better management of financial barriers (*p* = 0.037) suggests that educational attainment may offer some advantages in navigating these challenges, although financial barriers remained significant for many. This may be attributed to the fact that, despite their educational advantage, the majority of mothers included in this study are unemployed (83.5%), limiting financial resources and exacerbating financial barriers. This underscores the complexity of these barriers, indicating that financial and other access issues persist, even among families with stable employment or higher education levels [26]. Reducing treatment costs and improving insurance coverage have been shown to enhance access to dental services for SWDs, offering a potential solution to these financial challenges [6].

Furthermore, there was a statistically significant association (*p* = 0.048) between maternal education and difficulty with interior design barriers, with mothers reporting more difficulty with interior design barriers in dental facilities. This may reflect their role as primary caregivers and their heightened awareness of the physical environment’s impact on their SWDs’ comfort. This finding underscores the critical importance of adopting universal design principles in dental facilities to ensure these environments are both physically accessible and disability-friendly [31,32].

This suggests that financial stability alone is insufficient to overcome the challenges of accessing consistent and comprehensive dental care for persons with disabilities. International research supports this view, identifying a range of barriers, from physical accessibility to the availability of trained professionals [33], and emphasising the need for education, training, and increasing awareness about dental hygiene and annual dental check-ups to improve access to care [18].

While the findings provide valuable insights, it is important to acknowledge the limitations of this study. The findings may not be generalisable beyond the study region. The reliance on a non-random sampling method and self-reported data may introduce bias, and the cross-sectional design limits causal inference. The response rate of 49.9% was relatively high compared to similar studies [16,34]; however, it may still introduce some response bias. Additionally, the study does not capture the perspectives of oral health care providers, which could offer further insights into barriers to care for SWDs.

The study underscores the need for policies that promote training in managing SWDs and better integration of dental services with other health services to provide comprehensive care. Nowghani et al. demonstrated that multidisciplinary training initiatives designed for healthcare students from various disciplines, such as dental science, dental hygiene, speech and language therapy, and nursing, significantly improved self-efficacy and awareness of barriers to oral care for people with disabilities [35]. Additionally, healthcare policymakers can further improve oral health outcomes for SWDs by investing in cost-effective dental public health intervention strategies. These can be implemented through dental schools and providers in SWD schools, including school-supervised toothbrushing programs, pit and fissure sealants, fluoride varnish applications, mobile dental clinics, and tele-dentistry [36,37,38,39,40,41]. Mobile clinics deliver dental care directly at schools, while tele-dentistry enables remote consultations and follow-ups.

## 5. Conclusions

The findings highlight significant disparities in oral health behaviours and access to dental care among the study population. These results highlight the need for active collaboration between oral healthcare providers, educators, families, and health policymakers to address these challenges. Targeted educational interventions, particularly for mothers, could significantly enhance SWDs’ oral hygiene practices. Future policies should prioritise the implementation of effective preventive and interventional strategies, which can be strategically integrated through schools to improve the overall oral health of SWDs within the study region and beyond. Additionally, future researchers should evaluate the effectiveness of these initiatives and policy changes in improving oral health outcomes and reducing barriers to care among SWDs.

## Figures and Tables

**Figure 1 healthcare-12-01955-f001:**
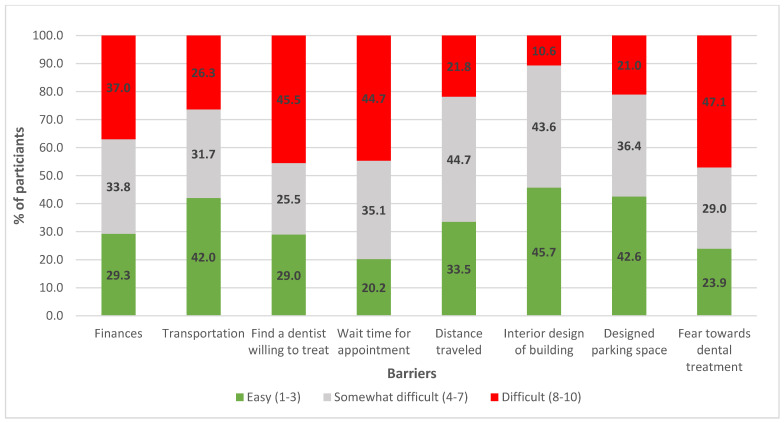
Barriers to dental care for students with disabilities: parents’ perspectives.

**Figure 2 healthcare-12-01955-f002:**
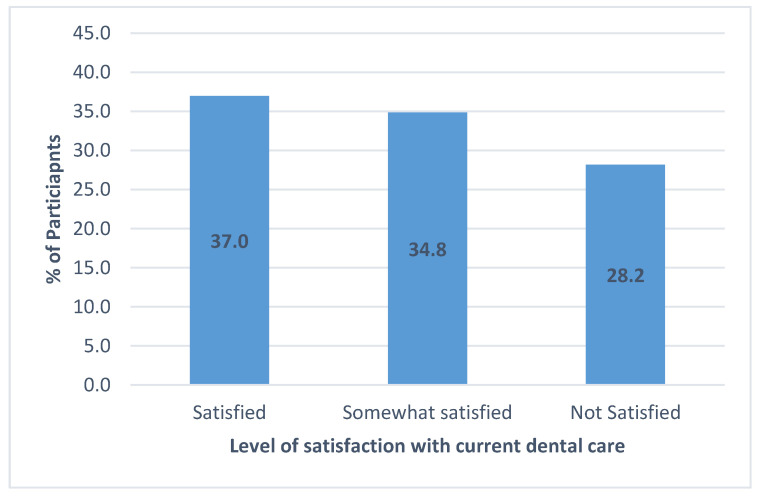
Parental attitudes and satisfaction with dental care for students with disabilities.

**Figure 3 healthcare-12-01955-f003:**
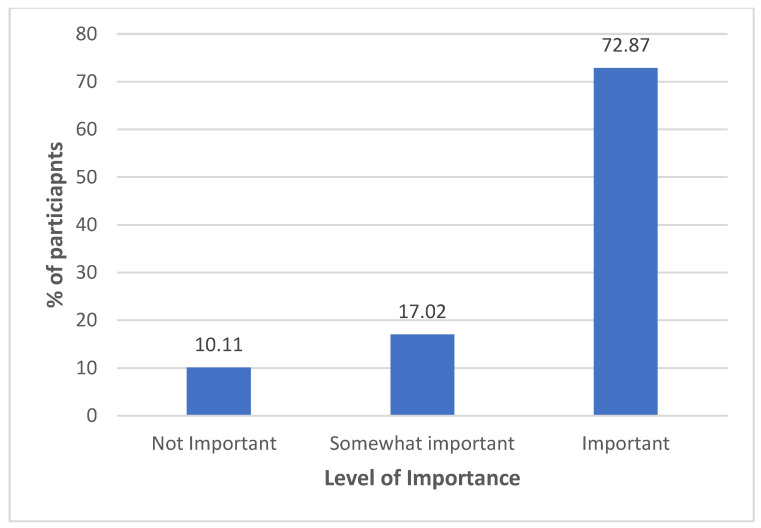
Importance of access to routine dental care.

**Table 1 healthcare-12-01955-t001:** Sociodemographic factors.

Variables	*N* (%)
Student’s gender	
Male	243 (64.6)
Female	133 (35.4)
Student’s age	
6–12	210 (55.9)
13–17	127 (33.8)
18–22	39 (10.4)
Relationship to the student:	
Mother	196 (52.1)
Father	151 (40.2)
Other (specify)	29 (7.7)
Mother/female guardian education level	
Illiterate	19 (5.1)
Elementary	52 (13.8)
Intermediate	60 (15.9)
High school	110 (29.3)
Above high school	135 (35.9)
Father/male guardian education	
Illiterate	18 (4.8)
Elementary	73 (19.4)
Intermediate	73 (19.4)
High school	108 (28.7)
Above high school	104 (27.7)
Maternal occupation	
Employed	62 (16.5)
Unemployed	314 (83.5)
Paternal occupation	
Employed	285 (75.8)
Unemployed	91 (24.2)
Student’s education level	
Elementary	244 (64.9)
Intermediate	64 (17)
High school	68 (18.1)
Student’s impairment condition according to school record	
Sensory	149 (39.6)
Cognitive	101 (26.9)
Autism Spectrum Disorder	99 (26.3)
Severe Multiple Impairment	27 (7.2)

**Table 2 healthcare-12-01955-t002:** Distribution of oral health behaviours and perceived oral health status.

Variables	*N* (%)
Hand coordination and ability to brush own teeth	
None	15 (3.9)
Severely limited	24 (6.4)
Able to do with some limitation	70 (18.6)
Able to do with no limitation	267 (71.0)
Tooth brushing	
Twice a day or more	63 (16.8)
Once a day	121 (32.2)
Irregular	172 (45.7)
Never	20 (5.3)
Dental flossing	
Yes	16 (4.3)
No	360 (95.7)
Sweets intake	
Never	13 (3.5)
Rarely in small quantities	107 (28.5)
Once or twice a day in small quantities	181 (48.1)
Once or twice a day in large quantities	42 (11.1)
More than twice a day in large quantities	33 (8.8)
Perceived oral health status	
Excellent	54 (14.4)
Good	151 (40.1)
Moderate	120 (31.9)
Poor	51 (13.6)

**Table 3 healthcare-12-01955-t003:** Oral health care utilisations.

Variables	*N* (%)
Frequency of annual dentist visits	
Three or more times a year	16 (4.3)
Twice a year	20 (5.3)
Once a year	34 (9.1)
Emergency basis only	195 (51.9)
Never	111 (29.5)
Past dental procedures	
Cleaning teeth	75 (19.9)
Filling tooth/teeth	97 (25.8)
Oral examination	78 (20.7)
Crown/bridge	1 (0.3)
Dentures (partial or complete)	4 (1.1)
Root canal therapy	23 (6.1)
Tooth extraction	110 (29.3)
Emergency treatment	58 (15.4)
Unknown	72 (19.2)
Received sedation for dental treatment	
Yes	103 (27.4)
No	273 (72.6)
If received sedation, type	
General anaesthesia in a hospital	42 (41)
IV conscious in a hospital or clinic setting	3 (2.9)
Oral medication for sedation only in a hospital or clinic setting	21 (20.3)
Minimal inhaled sedation (‘laughing gas’)	14 (13.5)
Not sure/do not remember	23 (22.3)
Type of facility for dental care	
Do not know	80 (21.3)
Hospital	87 (23.1)
Primary health centre	62 (16.5)
Private general practice	63 (16.8)
Paediatric general dental practice	84 (22.3)

**Table 4 healthcare-12-01955-t004:** Association between sociodemographic and oral health behaviours.

Sociodemographic Variables	Oral Health Behaviour Factors					*p*-Value
Frequency of tooth brushing						
	Twice a day or more	Once a day	Irregular	Never		
Mother’s education						
Illiterate	1 (1.6)	3 (2.5)	11 (6.4)	4 (20)		0.004
Elementary	12 (19.1)	18 (14.9)	16 (9.3)	6 (30)		
Intermediate	13 (20.6)	17 (14.1)	30 (17.4)	0		
High school	13 (20.6)	38 (31.4)	56 (32.6)	3 (15)		
Above high school	24 (38.1)	45 (37.2)	59 (34.3)	7 (35)		
Relationship to child						
Mother	39 (61.9)	61 (50.4)	88 (51.2)	8 (40)		0.693
Father	20 (31.8)	51 (42.2)	70 (40.7)	10 (50)		
Other	4 (6.4)	9 (7.4)	14 (8.1)	2 (10)		
Father’s education						
Illiterate	3 (4.8)	4 (3.3)	9 (5.2)	2 (10)		0.490
Elementary	13 (20.6)	19 (15.7)	38 (22.1)	3 (15)		
Intermediate	13 (20.6)	29 (23.9)	25 (14.5)	6 (30)		
High school	14 (22.2)	37 (30.6)	54 (31.4)	3 (15)		
Above high school	20 (31.8)	32 (26.5)	46 (26.7)	6 (30)		
Maternal occupation						
Employed	13 (20.6)	23 (19.0)	23 (13.4)	3 (15)		0.459
Unemployed	50 (79.4)	98 (80.9)	149 (86.6)	17 (85)		
Paternal occupation						
Employed	48 (76.2)	93 (76.9)	128 (74.4)	16 (80)		0.930
Unemployed	15 (23.8)	28 (23.1)	44 (25.6)	4 (20)		
Frequency of dental flossing						
	Yes	no				*p* value
Relationship to child						
Mother	14 (87.5)	182 (50.6)				0.012
Father	1 (6.3)	150 (41.7)				
Other (specify)	1 (6.3)	28 (7.8)				
Mother’s education						
Illiterate	1 (6.3)	18 (5)				0.689
Elementary	4 (25)	48 (13.3)				
Intermediate	2 (12.5)	58 (16.1)				
High school	5 (31.3)	105 (29.2)				
Above high school	4 (25)	131 (36.4)				
Father’s education						
Illiterate	1 (6.25)	17 (4.72)				0.672
Elementary	5 (31.25)	68 (18.89)				
Intermediate	2 (12.50)	71 (19.72)				
High school	3 (18.75)	105 (29.17)				
Above high school	5 (31.25)	99 (27.50)				
Maternal occupation						
Employed	5 (31.3)	57 (15.8)				0.104
Unemployed	11 (68.8)	303 (84.2)				
Paternal occupation						
Employed	10 (62.5)	275 (76.4)				0.204
Unemployed	6 (37.5)	85 (23.6)				
Frequency of sweets intake						
	Never	Rarely in small quantity	1–2×/day, small qty.	1–2×/day in large qty.	>2×/day in large qty.	*p*-value
Mother’s education						
Illiterate	0	7 (6.5)	6 (3.3)	3 (7.1)	3 (9.1)	0.036
Elementary	2 (15.4)	26 (24.3)	17 (9.4)	4 (9.5)	3 (9.1)	
Intermediate	2 (15.4)	14 (13.1)	31 (17.1)	4 (9.5)	9 (27.3)	
High school	3 (23.1)	31 (28.9)	53 (29.3)	12 (28.6)	11 (33.3)	
Above high school	6 (46.2)	29 (27.1)	74 (40.9)	19 (45.2)	7 (21.2)	
Paternal education						
Illiterate	0 (0)	7 (6.5)	7 (3.9)	1 (2.4)	3 (9.1)	0.04
Elementary	2 (15.4)	24 (22.4)	33 (18.2)	11 (26.2)	3 (9.1)	
Intermediate	2 (15.4)	18 (16.8)	46 (25.4)	4 (9.5)	3 (9.1)	
High school	2 (15.4)	36 (33.6)	42 (23.2)	13 (30.9)	15 (45.5)	
Above high school	7 (53.9)	22 (20.6)	53 (29.3)	13 (30.5)	9 (27.3)	
Maternal occupation						
Employed	3 (23.1)	18 (16.9)	25 (13.8)	12 (28.7)	4 (12.1)	0.180
Unemployed	10 (76.9)	89 (83.2)	156 (86.2)	30 (71.3)	29 (87.9)	
Paternal occupation						
Employed	11 (84.6)	79 (73.8)	136 (75.2)	32 (76.9)	27 (81.8)	0.831
Unemployed	2 (15.4)	28 (26.2)	45 (24.9)	10 (23.1)	6 (18.2)	
Relationship to child						
Mother	5 (38.5)	53 (49.5)	101 (55.8)	22 (52.8)	15 (45.5)	0.357
Father	7 (53.9)	40 (37.4)	71 (39.2)	18 (42.6)	15 (45.5)	
Other (specify)	1 (7.7)	14 (13.1)	9 (4.9)	2 (4.76)	3 (9.1)	

**Table 5 healthcare-12-01955-t005:** Association between sociodemographic and oral health utilisation.

Frequency Visiting a Dentist per Year						
	Three or More	Twice	Once	Emergency Basis Only	Never	*p*-Value
Father’s education						
Illiterate	0 (0)	0	0	11 (5.64)	7 (6.31)	0.035
Elementary	5 (31.3)	5 (25)	5 (14.7)	35 (17.9)	23 (20.7)	
Intermediate	1 (6.3)	2 (10)	7 (20.6)	39 (20)	24 (21.6)	
High school	7 (43.8)	7 (35)	3 (8.8)	57 (29.2)	34 (30.6)	
Above high school	3 (18.8)	6 (30)	19 (55.9)	53 (27.2)	23 (20.7)	
Mother’s education						
Illiterate	0	0 (0)	2 (5.9)	8 (4.1)	9 (8.1)	0.192
Elementary	2 (12.5)	1 (5)	4 (11.8)	27 (13.9)	18 (16.2)	
Intermediate	7 (43.8)	1 (5)	4 (11.8)	31 (15.9)	17 (15.3)	
High school	5 (31.3)	8 (40)	11 (32.4)	57 (29.2)	29 (26.1)	
Above high school	2 (12.5)	10 (50)	13 (38.2)	72 (36.9)	38 (34.2)	
Maternal occupation						
Employed	1 (6.3)	7 (35)	4 (11.8)	33 (16.9)	17 (15.3)	0.142
Unemployed	15 (93.8)	13 (65)	30 (88.2)	162 (83.1)	94 (84.7)	
Paternal occupation						
Employed	9 (56.25)	19 (95)	29 (85.29)	142 (72.82)	86 (77.48)	0.038
Unemployed	7 (43.75)	1 (5)	5 (14.71)	53 (27.18)	25 (22.52)	
Relationship to child						
Mother	8 (50)	12 (60)	13 (38.2)	105 (53.9)	58 (52.3)	0.419
Father	8 (50)	8 (40)	19 (55.9)	72 (36.9)	44 (39.6)	
Other (specify)	0	0	2 (5.88)	18 (9.23)	9 (8.11)	
Type of facility						
	Do not Know	Hospital	Primary Health Centre	Private General Practice	Paediatric Dental Practice	*p*-Value
Paternal occupation						
Employed	64 (80)	60 (68.9)	42 (67.7)	51 (80.9)	68 (80.9)	0.121
Unemployed	16 (20)	27 (31.0)	20 (32.3)	12 (19.1)	16 (19.1)	
Maternal occupation						
Employed	16 (20)	13 (14.9)	8 (12.9)	9 (14.3)	16 (19.1)	0.723
Unemployed	64 (80)	74 (85.1)	54 (87.1)	54 (85.7)	68 (80.9)	
Relationship to child						
Mother	41 (51.3)	42 (48.3)	26 (41.9)	38 (60.3)	49 (58.3)	0.426
Father	31 (38.8)	37 (42.5)	30 (48.4)	21 (33.3)	32 (38.1)	
Other (specify)	8 (10)	8 (9.20)	6 (9.7)	4 (6.4)	3 (3.6)	
Mother’s education						
Illiterate	7 (8.8)	3 (3.5)	5 (8.1)	3 (4.8)	1 (1.2)	0.144
Elementary	10 (12.5)	14 (16.1)	14 (22.6)	7 (11.1)	7 (8.3)	
Intermediate	14 (17.5)	10 (11.5)	11 (17.7)	11 (17.5)	14 (16.7)	
High school	20 (25)	31 (35.6)	18 (29)	14 (22.2)	27 (32.1)	
Above high school	29 (36.3)	29 (33.3)	14 (22.6)	28 (44.4)	35 (41.7)	
Father’s education						
Illiterate	6 (7.5)	4 (4.6)	2 (3.2)	2 (3.2)	4 (4.8)	0.772
Elementary	13 (16.3)	13 (14.9)	15 (24.2)	16 (25.4)	16 (19.1)	
Intermediate	16 (20)	17 (19.5)	13 (20.9)	15 (23.8)	12 (14.3)	
High school	22 (27.5)	30 (34.5)	19 (30.7)	13 (20.6)	24 (28.6)	
Above high school	23 (28.8)	23 (26.4)	13 (20.9)	17 (26.9)	28 (33.3)	

**Table 6 healthcare-12-01955-t006:** Association of barriers with sociodemographic variables.

Barrier	Variable	Easy (1–3)	Somewhat Difficult (4–7)	Difficult (8–10)	*p*-Value
Finance	Relationship to child				0.811
	Mother	60 (54.55)	69 (54.33)	67 (48.20)	
	Father	43 (39.09)	48 (37.80)	60 (43.17)	
	Other (specify)	7 (6.36)	10 (7.87)	12 (8.63)	
	Mother’s education				0.037
	Illiterate	2 (1.82)	5 (3.94)	12 (8.63)	
	Elementary	13 (11.82)	19 (14.96)	20 (14.39)	
	Intermediate	13 (11.82)	22 (17.32)	25 (17.99)	
	High school	29 (26.36)	37 (29.13)	44 (31.65)	
	Above high school	53 (48.18)	44 (34.65)	38 (27.34)	
	Father’s education				0.176
	Illiterate	2 (1.82)	6 (4.72)	10 (7.19)	
	Elementary	17 (15.45)	26 (20.47)	30 (21.58)	
	Intermediate	26 (23.64)	23 (18.11)	24 (17.27)	
	High school	26 (23.64)	39 (30.71)	43 (30.94)	
	Above high school	39 (35.45)	33 (25.98)	32 (23.02)	
	Maternal occupation				0.954
	Employed	19 (17.27)	21 (16.54)	22 (15.83)	
	Unemployed	91 (82.73)	106 (83.46)	117 (84.17)	
	Paternal occupation				0.008
	Employed	88 (80)	104 (81.89)	93 (66.91)	
	Unemployed	22 (20)	23 (18.11)	46 (33.09)	
Interior Design	Relationship to child				0.048
	Mother	90 (52.33)	87 (53.05)	19 (47.50)	
	Father	63 (36.63)	72 (43.90)	16 (40)	
	Other (specify)	19 (11.05)	5 (3.05)	5 (12.50)	
	Mother’s education				0.473
	Illiterate	6 (3.49)	10 (6.10)	3 (7.50)	
	Elementary	24 (13.95)	19 (11.59)	9 (22.50)	
	Intermediate	24 (13.95)	29 (17.68)	7 (17.50)	
	High school	55 (31.98)	48 (29.27)	7 (17.50)	
	Above high school	63 (36.63)	58 (35.37)	14 (35)	
	Father’s education				0.009
	Illiterate	5 (2.91)	8 (4.88)	5 (12.50)	
	Elementary	30 (17.44)	28 (17.07)	15 (37.50)	
	Intermediate	39 (22.67)	30 (18.29)	4 (10)	
	High school	46 (26.74)	52 (31.71)	10 (25)	
	Above high school	52 (30.23)	46 (28.05)	6 (15)	
	Maternal occupation				0.555
	Employed	27 (15.70)	26 (15.85)	9 (22.50)	
	Unemployed	145 (84.30)	138 (84.15)	31 (77.50)	
	Paternal occupation				0.138
	Employed	138 (80.23)	120 (73.17)	27 (67.50)	
	Unemployed	34 (19.77)	44 (26.83)	13 (32.50)	

**Table 7 healthcare-12-01955-t007:** Association between sociodemographic variables and perceived importance of routine dental care for SWDs.

How Important Is Your Access to Routine Dental Care	Not Important (1–3)	Somewhat Important (4–7)	Important (8–10)	*p*-Value
Relationship to child				
Mother	20 (52.63)	31 (48.44)	145 (52.92)	0.836
Father	16 (42.11)	29 (45.31)	106 (38.69)	
Other (specify)	2 (5.26)	4 (6.25)	23 (8.39)	
Mother’s education				
Illiterate	0 (0)	1 (1.56)	18 (6.57)	0.235
Elementary	5 (13.16)	8 (12.50)	39 (14.23)	
Intermediate	8 (21.05)	11 (17.19)	41 (14.96)	
High school	10 (26.32)	26 (40.62)	74 (27.01)	
Above high school	15 (39.47)	18 (28.12)	102 (37.23)	
Father’s education				
Illiterate	1 (2.63)	0 (0)	17 (6.20)	0.041
Elementary	9 (23.68)	15 (23.44)	49 (17.88)	
Intermediate	13 (34.21)	14 (21.88)	46 (16.79)	
High school	8 (21.05)	22 (34.38)	78 (28.47)	
Above high school	7 (18.42)	13 (20.31)	84 (30.66)	
Maternal occupation				
Employed	3 (7.89)	12 (18.75)	47 (17.15)	0.307
Unemployed	35 (92.11)	52 (81.25)	227 (82.85)	
Paternal occupation				
Employed	3 (7.89)	12 (18.75)	47 (17.15)	0.307
Unemployed	35 (92.11)	52 (81.25)	227 (82.85)	

## Data Availability

The data that support the findings of this study are available from the corresponding author upon reasonable request.

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
