# Peer review of "Oral Health Behaviour, Care Utilisation, and Barriers among Students with Disabilities: A Parental Perspective"

_healthcare, 2024, doi:10.3390/healthcare12191955_

Round 1
Reviewer 1 Report
Comments and Suggestions for Authors
Dear Authors,
The paper was really well writeen, but I missed the conclusions section.
Author Response
Thank you for your positive feedback on the manuscript and for pointing out the need for a conclusions section. In response to your suggestion, we have included the following conclusions section:
Conclusion:
The findings highlight significant disparities in oral health behaviours and access to dental care among vulnerable students with disabilities (SWDs) in the study population. These results emphasise the need for active collaboration between oral healthcare providers, educators, and families to address these challenges. Future policies should prioritise the implementation of effective preventive and interventional strategies, which can be strategically integrated through schools to improve the overall oral health of this vulnerable population, both within the study region and beyond.
Reviewer 2 Report
Comments and Suggestions for Authors
hello authors
the topic is of interest, but can be improved further
- the introduction looks to be simple, we ask to address all the aspects related to the title of the topic and address the short comings in the introduction why this research was planned
- can you add the details how you calculated the sample size
- what the authors address related to barriers faced how they suggest to overcome them . were the barriers related to parents/ child or both
Comments on the Quality of English Language
English revision required for improving the quality of the article
Author Response
Thank you for your positive feedback and for recognizing the importance of our topic. We appreciate your suggestions and have made the following revision
- Introduction: We have enhanced the introduction and reordered the sentences in line with feedback from other reviewers. The core points of the topic have been highlighted more concisely, while a detailed discussion on key aspects has been introduced in the Discussion section.
- Sample Size Calculation: In response to your comment, we have added detailed information about the sample size calculation in the Methodology section. The sample size was calculated using an online calculator (Raosoft), based on the expected population of 4344 students with disabilities (SWDs) in Al-Ahsa, Saudi Arabia. We aimed for a 95% confidence level and a 5% margin of error, which resulted in a minimum required sample size of 354 participants. You can access the calculator at this link: http://www.raosoft.com/samplesize.html.
- Barriers and Solutions: The barriers identified were reported by parents/caregivers and primarily related to the students with disabilities. The most frequently reported barriers were fear of treatment (47.1%), difficulty in finding a dentist willing to treat the child (45.5%), and long wait times for appointments (44.7%). These barriers directly affect both the students and their families, as students with disabilities rely on their caregivers to access dental care. In the Discussion section, we suggest several solutions to overcome these challenges:
- Training for Dental Professionals: There remains a critical need for education and training for both current and future dentists to enhance their preparedness and attitudes in managing SWDs, which could ultimately lead to improvements in dental care access for SWDs. Nowghani et al. demonstrated that multidisciplinary training initiatives designed for healthcare students from various disciplines, such as dental science, dental hygiene, speech and language therapy, and nursing, significantly improved self-efficacy and awareness of barriers to oral care for people with disabilities.
- Sensory-Friendly Dental Environments: We recommend the use of sensory adaptations such as dimmed lighting, calming music, weighted blankets, and tactile familiarization. Additionally, assistive technologies like hearing aids, sign language interpreters, and braille materials should be incorporated to help children with disabilities feel more comfortable during their dental visits.
- Improved Accessibility: We also highlight the importance of adopting universal design principles in dental facilities to address physical accessibility issues reported by parents, particularly regarding the interior design of dental facilities.
- Financial Support and Policy Changes: While financial barriers were less frequently cited, they remain significant for some families. Reducing treatment costs and improving insurance coverage have been shown to enhance access to dental services for SWDs, offering a potential solution to these financial challenges.
- Public Health Programs: We propose the implementation of public health strategies such as mobile dental clinics and teledentistry services to bring dental care directly to schools and remote areas. These strategies can provide cost-effective preventive care, such as fluoride varnish applications and supervised toothbrushing programs in school settings. Targeted educational interventions, particularly for mothers, could significantly enhance SWDs' oral hygiene practices
We hope these revisions meet your expectations, and we appreciate your thoughtful insights, which have contributed to enhancing the manuscript.
Reviewer 3 Report
Comments and Suggestions for Authors
The study aims to explore oral health behaviours, dental care utilisation and barriers to accessing dental care among students with disabilities (SWDs) in Saudi Arabia, as reported by their parents. A descriptive cross-sectional analysis of sociodemographic factors & oral health practices & access to care was made.
The research is interesting and important as questions and problems presented are actual worldwide. The article is well written and needing only minor revision. Congratulations to the authors on their work in the special care dentistry field!
Major remark
The research must be repeatable, so it is necessary to put the questionnaire in the manuscript.
Minor remarks
Materials and Methods
It is necessary to describe study group and methods in this section and not in the Results. Did you adhere to the calculated sample size? How many schools were included? What was the age of students?
Line 104: “Each school independently identified eligible students with the assistance of teachers and staff…”. It is not clear what was the criteria for inclusion/exclusion of “eligible” students. It is a bit clearer if a reader goes back to the Line 81 “We selected participants through convenience sampling, 81 focusing on their accessibility and willingness to participate.” Sampling must be described clearly in the dedicated section of the Material & Methods.
Results
Authors discuss "regular flossing" but Table 2 does not contain that category, just yes/no answer. Please give information what is considered regular flossing and change categories in the table and throughout manuscript.
Sweets consumption - please define precisely what is "rarely".
Discussion
One of the conclusions is: “Establishing specialist training initiatives for dentists is imperative”, line 314. Isn't it more important to educate dental students so all general dentists in primary care in the future are educated and prepared for patients with disabilities. When patients have to seek help by specialists only, it is a barrier.
Please see more comments in the attached copy of the manuscript.

Comments on the Quality of English LanguageOnly minor language edits are necessary which are marked in the comments in the attached document.
Author Response
Thank you for your encouraging feedback and for highlighting the importance of our work. We appreciate your thoughtful comments and have made the following revisions to address your concerns:
Major Remark:
- Inclusion of the Questionnaire:
In response to your suggestion for ensuring the repeatability of the research, we have added the full questionnaire to the supplementary material of the manuscript.
Minor Remarks:
- Materials and Methods:
- Study Group and Methods:
We have moved the description of the study group and methods to the Materials and Methods section for clarity. This section now includes details on the sample size (376 participants), the number of schools involved (21), and the age range of students (6–22 years). We adhered to the calculated sample size, which was 354 participants, and exceeded it by including 376 participants. - Criteria for Eligibility:
We have clarified the eligibility criteria, which included students with disabilities who were officially registered as SWDs in the education system in Al-Ahsa, Saudi Arabia, and whose parents provided informed consent​. These revisions are now reflected in the Materials and Methods section.
Results:
‘’Authors discuss "regular flossing" but Table 2 does not contain that category, just yes/no answer. Please give information what is considered regular flossing and change categories in the table and throughout manuscript.’’
Thank you for pointing this out. This was an oversight, and the manuscript has been updated accordingly to reflect that the flossing category was a yes/no question.
‘’Sweets consumption - please define precisely what is "rarely".’’
Thank you for your observation regarding the need for more clarity in defining "rarely." The questionnaire uses specific categories for daily sweets consumption: "1–2×/day in small quantity," "1–2×/day in large quantity," and ">2×/day in large quantity." Given that these categories describe daily consumption, "rarely in small quantity" refers to consumption that occurs less frequently than once per day. This could indicate occasional consumption, such as once or twice per week. We have now clarified this definition in the manuscript to ensure consistency in the Results section.
Thank you for your thoughtful remarks and the opportunity to clarify our intent. Regarding the point on specialist training initiatives (line 314), we agree that educating dental students is crucial for preparing all general dentists in primary care to effectively manage patients with disabilities. Our intention with the phrase "specialist training initiatives" was to highlight the need for interdisciplinary education. In particular, we referenced Nowghani et al., who demonstrated that multidisciplinary training programs designed for healthcare students from various fields, including dental science, dental hygiene, and nursing, significantly improved self-efficacy and awareness of the barriers to oral care for people with disabilities. We have revised the text to enhance clarity and our intended meaning.
Thank you for your valuable feedback and the detailed comments provided in the attached copy of the manuscript. We have carefully reviewed and addressed all of your comments in the manuscript. Kind regards
Reviewer 4 Report
Comments and Suggestions for Authors
Thank you for the opportunity to consider this interesting work.
The manuscript submitted for potential publication in Healthcare aims to explore oral health behaviours, dental care utilisation, and barriers to accessing dental care among SWDs in Al-Ahsa, Saudi Arabia, from the perspectives of their parents.
Dear Authors,
You analyse very important aspects of oral health. However, there are a number of improvements which are required before it can be accepted.
Who is the subject of the study? Uniform naming in the workplace would be necessary, as children are not persons over 18 years of age.
1. Introduction
Concise text, a good introduction to the topic. If you could, it would be perfect to rewrite the introduction a bit to avoid so many repetitions of phrases: for example oral health.
I would remove this part: line 34, „for instance, about 16% of the global population…”, as I do not understand the correlation between the first part of the sentence and the example.
Suggestion- ( line 44) To move „Children and young people with special educational needs and disabilities often face significant challenges in maintaining good oral hygiene.”- after( line 34) „…various personal and environmental factors;”
2. Aim of the work
Line 60 ” The findings highlight the disparities in oral health care access for SWDs within the Saudi population and offer insights into the broader challenges faced by this group.” Please remove the sentence - it is not proper for aim section. You could move it to conclusions
3. Methodology
*Line 65 “We aimed in this study..”- please remove it. It is a repetition.
*Line 97 Pre-testing on 10 parents with 354 as sample size seems not adequate.
*I can not find the definition of the student in the methodology. What age are the people being tested? Are they children? Adults?
4. Specify in Table 1- gender and sex. Gender of student? Age of student?
5. Result section
Line 123 - Children are not aged 18-22. Please, correct it throughout the whole manuscript.
Line 134- Sugar intake is described in the Table, so in the text, there is no need to repeat the Table. It would be good to underline the most important aspects in the text.
Table 4- Frequency of dental flossing- Is there a sense to analyse it, as students do not floss?
Tables 4,5 and 6 are very long. Maybe figures would be more accessible for readers?
7. Conclusions
This section would be a very good summary of the manuscript.
Author Response
Thank you for your valuable feedback and for highlighting areas of improvement. We have carefully considered your suggestions and made the necessary revisions to the manuscript. Below is a point-by-point response to your comments:
- Introduction:
- We have revised the introduction to reduce the repetition of phrases, particularly "oral health," while maintaining clarity and flow.
- The part in line 34, “for instance, about 16% of the global population...,” has been removed as per your suggestion, as the correlation between the two parts of the sentence was unclear.
- We have moved the sentence "Children and young people with special educational needs and disabilities often face significant challenges in maintaining good oral hygiene" (line 44) to follow "various personal and environmental factors" (line 34) to improve readability and logical flow.
- Aim of the Work:
- We have removed the sentence, "The findings highlight the disparities in oral health care access for SWDs within the Saudi population and offer insights into the broader challenges faced by this group" from the aim section and relocated it to the conclusions, where it is more appropriate.
- Methodology:
- The sentence "We aimed in this study..." in line 65 has been removed as it was redundant and repeated.
- Thank you for your comment. The questionnaire was developed by modifying and adding items to a previously published survey, and these modifications were reviewed by experts to ensure relevance and comprehensiveness. It was then pre-tested in two phases: first with 5 male and 5 female parents to check for item inclusion and wording clarity, followed by a second pilot test with 10 parents to confirm the final version’s clarity and effectiveness. This process ensured that the questionnaire was clear and comprehensive. The pre-testing was not related to the overall sample size of 354 participants but focused specifically on items and wording clarity.
- We have added a clear definition of the participants in the study, specifying the age range of students (6–22 years) and whether they are children or adults. The manuscript now refers to these participants as students with disabilities (SWDs) instead of children, and this distinction has been consistently applied throughout the manuscript.
- Table 1:
- We have clarified the gender and age of the students in Table 1
- Results Section:
- regarding concern reference to students aged 18–22. The manuscript now refers to all as students with disabilities (SWDs) instead of children, and this distinction has been consistently applied throughout the manuscript.
- We have emphasized the key findings of information from the tables in the text, regarding the sugar intake described in Table 2.
- Regarding the analysis of dental flossing in Table 4: Although students infrequently reported flossing, we retained the analysis because it highlights an important oral hygiene behavior that needs improvement. Our study found a statistically significant association between the relationship to the SWDs and flossing, with mothers being more likely to report that SWDs floss. Despite this, the overall rate of flossing among SWDs remains very low (4.3%). This finding highlights that while mothers might be more involved in promoting flossing, the practice itself is rarely adopted- We have clarified this in the discussion.
- Regarding Tables 4, 5, and 6: These tables represent associations between variables, which are more clearly presented in tabular format due to the inclusion of statistical values such as p-values and measures of association. Converting these tables into graphs would reduce the clarity and interpretability of the data. Therefore, we have retained the tabular format to ensure accuracy and ease of understanding. Conclusion: We have added a conclusion section to provide a comprehensive summary of the key findings and the broader implications of our study Thank you and kind regards.